# Root Breeding in the Post-Genomics Era: From Concept to Practice in Apple

**DOI:** 10.3390/plants11111408

**Published:** 2022-05-26

**Authors:** Zhou Zhou, Lei Zhang, Jing Shu, Mengyu Wang, Han Li, Huairui Shu, Xiaoyun Wang, Qinghua Sun, Shizhong Zhang

**Affiliations:** 1State Key Laboratory of Crop Biology, Shandong Agricultural University, Tai’an 271018, China; zhouzhou2021@126.com (Z.Z.); zhanganqi2002@163.com (L.Z.); wmengyu2021@163.com (M.W.); lihan@sdau.edu.cn (H.L.); hrshu@sdau.edu.cn (H.S.); xyunwang@sdau.edu.cn (X.W.); 2College of Forestry Engineering, Shandong Agriculture and Engineering University, Jinan 250100, China; shjing79@163.com

**Keywords:** apple germplasm resources, genomics, precise breeding

## Abstract

The development of rootstocks with a high-quality dwarf-type root system is a popular research topic in the apple industry. However, the precise breeding of rootstocks is still challenging, mainly because the root system is buried deep underground, roots have a complex life cycle, and research on root architecture has progressed slowly. This paper describes ideas for the precise breeding and domestication of wild apple resources and the application of key genes. The primary goal of this research is to combine the existing rootstock resources with molecular breeding and summarize the methods of precision breeding. Here, we reviewed the existing rootstock germplasm, high-quality genome, and genetic resources available to explain how wild resources might be used in modern breeding. In particular, we proposed the ‘from genotype to phenotype’ theory and summarized the difficulties in future breeding processes. Lastly, the genetics governing root diversity and associated regulatory mechanisms were elaborated on to optimize the precise breeding of rootstocks.

## 1. Introduction

With their distinct taste and rich nutrient contents, apple (*Malus domestica Borkh.*) fruits are widely enjoyed and have become popular among consumers. Grafting is used in apple production to ensure the desired characteristics of fruit trees in orchards, shorten the lengthy juvenile period, and ensure high fruit quality and high yields [1,2]. Rootstocks, which are a component of grafted plants found mostly belowground, advance the fruiting period of apple scion varieties, affect flowering and fruit set and exert important effects on the growth and development of scion varieties, yield, quality, and their resistance to various natural disasters [3]. An important component of rootstock breeding is to select a more developed and structured root system based on the selection of dwarf rootstocks. However, the roots of rootstocks are buried deep underground, and many limitations are encountered when only relying on past planting experience to select and breed rootstock varieties with excellent varieties, strong stress resistance and strong root system configurations. Therefore, the utilization of modern biological technology to improve breeding and achieve the purpose of the precise selection of apple rootstocks is an urgent need. In recent years, whole-genome sequencing has been conducted on a large number of apple varieties and provides biological information for the molecular design and breeding strategies of fruit tree species [4]. In particular, apple genomic research has been conducted on the basis of the success of the human, *Arabidopsis*, rice and other model organism genome projects. Advances in apple genomic research have substantially promoted the generation of the whole-genome sequence of related fruit tree species, the construction of apple database resources, the determination of the origin and evolution of apple species, and the determination of the location and cloning of apple functional genes [5,6]. Following the establishment of the apple genome database and partnered with molecular marker breeding, transgenic breeding, molecular design breeding and other methods, the combination of genotypic and phenotypic information has become an important contribution to methods for species improvement. Molecular breeding has the advantages of shortening the breeding period, improving individual traits, and overcoming outbreeding incompatibility. Therefore, molecular breeding has become an indispensable component of breeding efforts [7]. Previously, empirical breeding and cross-breeding work required decades of resource accumulation in the selection of the most suitable apple rootstocks, and large amounts of time and effort were necessary to evaluate the capabilities and limitations of each rootstock under each growing condition. Therefore, the rapid development of plant genomics revolutionized apple breeding. In this paper, we provide a systematic survey of the genetic diversity of apple rootstocks among varieties, information on genome sequencing and the functional annotation of different varieties of apple as well as on the development of molecular markers and a summary of functional genes. The molecular network involved in stress resistance, nutrient regulation, hormone regulation and root configuration regulation in apple are also reviewed. On this basis, a new apple root response network was constructed and is presented.

In this review, several databases, including Elsevier, PubMed, SpringerLink, and Google Scholar, were used for the literature search. A combination of keywords such as apple germplasm resources, genomics, precise breeding, abiotic stress, biotic stress, drought, salt, temperature stress, nutrient regulation, and hormone regulation were used as search terms. Publications from 2010–2021 were preferred, but this range was extended in some cases. After searching the literature, we read each paper carefully and thoroughly to exclude those that did not meet the inclusion criteria. We initially selected 160 papers related to these targets.

## 2. Rootstock Genetic and Molecular Regulation

### 2.1. Abundance of Rootstock Resources and Genomic Diversity

Wild apple resources are often used as rootstocks due to their strong tolerance to harsh environments. Wild apple resources are widely distributed worldwide, with the following five distinct gene centers: East Asia, Central Asia, West Asia, Europe, and North America [8]. The main wild species of apple worldwide are classified into six groups, eight lines, twenty seven species, six subspecies, fourteen varieties, and three forms, while the cultivated species include eight species, one subspecies, and seven varieties [9,10]. Phenotypic or genetic differences between the same species are often referred to as varieties. For example, two varieties of *Begonia*, i.e., *Alba plena* and *Riverside*, were identified. In contrast, *Malus halliana* includes the varieties *parkmani* and *spontanea*, which are cultivated and formed during production [11]. Due to the complex geographic environment of rootstock growth and its distribution, as well as human-mediated interspecific and natural hybridization, rootstock varieties produced in different regions have different adaptabilities to the environment. For instance, *Malus baccata* is widely used as a rootstock and breeding resource in high-latitude apple production areas, and its cold resistance and zinc storage are significantly better than those of most rootstocks [12,13]. In addition, during the evolution of the genus *Malus*, various modes of reproduction promoted the diversity of apple varieties. Another method of reproduction, apomixis, occurs in wild apple species; apomixis is facultative to balancing the apomixis and sexual seeds [14]. In general, the abundance of wild apple resources constitutes the foundation of the genetic diversity of apple domestication.

The cultivation of rootstocks with strong environmental adaptability, compatibility, and reproductive ability represents an important and timely research direction for breeders. Wild rootstocks have strong resistance and well-developed root systems, but shortcomings such as tall trees and the difficulty of planting and forming flowers have become the main factors restricting the development of wild rootstocks. The selection and breeding of rootstocks may compensate for the lack of wild resources to a certain extent. In 1917, the British NIAB East Malling Research program initiated the Rootstock Project, which involved selecting and breeding M and MM (apple rootstock varieties that are widely used in production) rootstocks that remain influential to this day [15]. In the United States, a country in which the selection and breeding of apple rootstocks started earlier, an ongoing apple rootstock breeding program involving fire blight (the causal agent of which is Erwinia amylovora) resistance, neck rot resistance (caused by Phytophthora cactorum (Leb. et Cohn.) Schrot), and resistance to apple aphids since 1968 investigates the selecting and breeding of G rootstocks [16]. Breeding produced rootstock variety G.210, a semidwarfing rootstock that is resistant to fire blight and crown rot (the causal agent of which is Phytophthora cactorum) [17]. The MM series (United Kingdom), P series (Poland), B series (Soviet Union), CG series (United States), O series (Canada), A series (Sweden), JM series (Japan), and MAC series (United States) are all excellent germplasm resources [18]. However, the breeding history of modern rootstocks is relatively short, which has resulted in relatively few domesticated rootstock varieties and has narrowed the genetic diversity. The apple breeding cycle is too long, rootstock root system research is difficult, and systematic research on rootstock resources is still lacking. The existing rootstock varieties must be improved to adapt to the environment and production diversity. With the further development of sequencing technology and molecular biology, genome sequence assembly and resequencing studies have been conducted for an increasing number of species, resulting in a substantial increase in the amount of data obtained from these materials. As a result, research on each gene and each sequence has become more in-depth and detailed, which is useful for researchers when developing an understanding of biological problems more generally [19]. However, related studies on the apple genome using high-throughput sequencing have just begun, and the advantages of disease resistance and stress resistance have not been fully utilized in breeding work. In fact, high-throughput sequencing methods to analyze the genomic characteristics of apple wild resources and dwarf rootstocks and specific genes to improve existing dwarf rootstock resources for genome-directed breeding are critical for the genetic improvement of apple rootstocks.

In recent years, high-throughput genotyping and precise phenotyping have been widely used to screen germplasms for potential key allelic variants that affect traits, and the germplasm allele bank has been aided by modern technology for the genetic improvement of apple [20]. High-quality reference genomes are powerful tools for research on gene function, population evolution, and crop plant breeding. With the continuous improvement of the apple genome and in-depth analysis of the heterogeneous genetic diversity of germplasms, genome-assisted breeding (GAB) has become very convenient [21,22,23]. In 2010, the Malus × domestica Borkh. genome was sequenced for the first time, which served as a complete apple reference genome [24,25]. In 2017, by reassembling the Malus × domestica Borkh. genome, researchers constructed a more complete reference sequence [26,27]. In 2019, the chromosome assembly of the Hanfu (Dongguang × Fuji) apple HTFH1 genotype was clearly parsed, which promoted research on new genes and the genetic breeding of apple [28]. Whole-genome resequencing and population genetic research of nearly 400 cultivated apple genotypes, cultivated apple genotype relatives and major wild resources of the apple genus enhanced the understanding of the molecular evolution of this species [29,30]. The development of new apple reference genomes through sequencing platforms, especially the genomes of wild ancestors, will further help to understand the genetic diversity of apple and cultivate improved varieties of apple rootstocks. Therefore, methods to shift from the genome to phenotype become present an interesting area of research. The full use of genomics research results is necessary to improve the efficiency of the process of germplasm innovation and to discover new genes and favorable alleles present in germplasm resources.

Driven by genome sequencing, both the research and application of molecular markers have developed rapidly. Molecular marker-assisted selection based on whole-genome sequencing can shorten the breeding cycle and improve breeding efficiency [31]. The identification of molecular markers is the basis of breeding via molecular marker-assisted selection. Simple sequence repeats (SSRs) are one of the most effective genetic markers [32,33,34]. Researchers used nine SSR markers and four chloroplast markers to analyze the heterozygosity, genetic diversity, and genetic structure of 56 apple varieties from five different natural *Malus sylvestris* populations in Saxony, Germany [35]. In addition, the genetic diversity of 68 apple varieties was analyzed using 10 pairs of SSR primers and self-incompatibility sites, revealing the genetic relationships between apple varieties [36]. High-density maps based on genome sequencing provide a solid foundation for locating important trait-related quantitative trait loci (QTLs) [37,38]. QTLs and genome-wide linkage maps have also been used to analyze the genetic basis of fire blight disease in Rosaceae [39,40,41,42,43]. At present, effective high-density genetic maps for fine mapping and marker-assisted breeding in apple represent applications of high-quality genomes. By constructing a high-density genetic linkage map comprising the F1 population resulting from Red × Changfu No. 2, researchers identified potential QTL sites using the interval mapping method, and a multiple QTL model was established to identify the sites that control the anthocyanins of red meat apples [44]. Using QTL mapping and bulked segregant analyses to identify 10 QTLs related to fruit color through sequencing, researchers predicted 10 genes and established a nonadditive QTL-based genomics-assisted prediction model. These results are conducive to apple GAB and may provide insights for understanding the underlying mechanism [45]. Genomics can be applied to explore apple resources, identify key genes related to nutrient regulation and resistance of rootstocks, perform target transformation, and breed high-quality varieties.

### 2.2. Biotic and Abiotic Stress Modulation in Apple

The phenotype of an individual is the result of the combined effect of the genotype and environment. The main task of molecular breeding in apple is to identify genes that control target traits and aggregate favorable genes present in different materials to develop suitable varieties for agricultural production. The stress resistance of roots is the most critical trait affecting the growth of apple shoots. In the selection process of breeding, we should focus on root traits that allow cultivated varieties to grow well under biotic and abiotic stresses. At present, the molecular mechanisms underlying the biotic and abiotic stress resistance of apple are being gradually understood, which provides important references for the use of molecular breeding to develop rootstocks with good root quality.

Apples are constantly challenged by biotic and abiotic stresses during planting, including biotic stresses such as viruses, bacteria, fungi, herbivores, and parasitic plants, as well as abiotic stresses such as drought, salt, and cold. Apples have correspondingly evolved powerful defense mechanisms against the complex environment to adapt. Pathogens that infect plants produce a series of pathogenic proteins designed to specifically destroy the plant’s immune system, while plants have evolved a series of disease resistance proteins that rapidly activate the disease resistance response to limit and kill pathogens [46]. At the beginning of immunity, plants have both self- and non–self-recognition abilities. Therefore, pattern recognition receptors (PRRs) are one of the most basic components of plant immunity [47]. When pathogenic microorganisms infect plants, PRRs on the surface of plant cells recognize microbial PAMPs/MAMPs (microbe- or pathogen-associated molecular patterns), and the plants respond via pattern-triggered immunity (PTI) [48]. For instance, CHITIN ELICITOR RECEPTOR KINASE 1 (MdCERK1-2), a PRR, is induced and expressed by pathogenic bacteria and participates in apple antifungal defense by recognizing chitin and initiating the chitin-mediated immune response [49,50]. Lysin motif-containing protein (LYP) family members were identified in Rosaceae; LYPs are important components of PRRs, among which LYP1b2 and 4/5a1 are located in the plasma membrane, and most LYP genes show differential expression in leaves infected with fungal pathogens [51]. Elongation factor thermal unstable (EF-Tu) is one of the most abundant bacterial proteins and functions as a very potent PAMP in *Arabidopsis thaliana*. The ectopic expression of *Arabidopsis* EF-Tu in apple rootstock M.26 significantly enhanced its resistance to the bacterium *Erwinia amylovora* [52]. Recognition and immune regulation during the plant response to biotic stress have been extensively studied, but the hypotheses that have been proposed in recent years must be further verified in apple.

The conditions of apple orchards are generally poor, and most of them are located in areas with poor soil, saline-alkaline soil and poor water conditions. As a result, abiotic stress has become the main limiting factor restricting the growth and development of apple trees, which seriously affects the sustainable development of the apple industry [53]. Therefore, designing methods to enhance the ability of apple trees to resist abiotic stress and improve apple fruit yield and quality is an important responsibility of scientific and technological workers. Drought disrupts the normal growth state and physiology of apple. Therefore, apple species have evolved many mechanisms to adapt to drought. The transmission and transduction of drought signals by plants are regulated by multiple pathways, which are divided into the following two pathways: abscisic acid (ABA)-dependent and ABA-independent pathways [54]. Research on the relationships between the core ABA complex and other known regulatory pathways of stress signaling in plants is gradually being conducted for apple. The expression of ECERIFERUM 2 (*MdCER2*), *MhYTP1* and its homolog YT521 (*MhYTP2*), β-ketoacyl-CoA synthase 21 (*MdKCS21*), 14-3-3 protein 11 (*MdGRF11*), and cystatin gene 4 (*MpCYS4*) is induced in response to ABA and drought [55,56,57,58,59,60,61]. In addition, some members of the transcription factor (TF) families, such as the v-myb avian myeloblastosis viral oncogene homolog (MYB), B-box protein (BBX), APETALA 2/ETHYLENE RESPONSIVE FACTOR (AP2/ERF), basic helix-loop-helix (bHLH) protein, nuclear factor Y (NF-Y), major latex protein (MLP), WRKY, and homeobox families, have also been reported to participate in the drought response through the ABA signaling pathway [62,63,64,65,66,67]. For example, under drought stress, MdWRKY31 binds to the promoter of MdRAV1 to inhibit its transcription. Furthermore, MdRAV1 inhibits the expression of *abscisic acid-insensitive 3* (*MdABI3*) and *abscisic acid-insensitive 4* (*MdABI4*) [68] (Figure 1). In the MAPK cascade, MdRaf5, a member of the MAPKKK family, is upregulated in different apple rootstock varieties after treatment with polyethylene glycol (PEG). The heterologous transformation of *Arabidopsis* enhances the drought resistance of plants through the ABA pathway [69]. The MdMEK2-MdMPK6 cascade activated by water deficits regulates the MdWRKY17-MdSUFB (sulfur mobilization B) pathway and balances the survival and yield of apple plants under moderate drought stress by inhibiting chlorophyll degradation [70] (Figure 1). In addition, under drought conditions, flavonoids (especially anthocyanins) are synthesized in large quantities, and thus anthocyanins generally improve the drought resistance of plants. Flavonoids have a strong antioxidant capacity, the ability to remove reactive oxygen species (ROS) and have been widely reported to participate in the interaction between plants and the environment. Under drought stress, the MdHSP90 (heat shock protein)-MdHSFA8a (heat shock factor A8a) complex separates, and the released MdHSFA8a subsequently interacts with MdRAP2.12-related APETALA 2/ETHYLENE RESPONSIVE FACTOR (AP2/ERF) family TFs to activate flavonoid synthesis pathway-related genes and induce the synthesis of flavonoids [71] (Figure 1). By accelerating the degradation of the MdERF38 protein, MdBT2 reduces anthocyanin biosynthesis induced by MdERF38. Transgenic apple plants in which MdBT2 is knocked down have obvious stress-resistant phenotypes [72]. Key genes such as *MdATG18* and *MdRAV* are also involved in the pathway of anthocyanin synthesis and play an important role [73,74].

Excessive Na^+^ caused by salt stress may lead to a lack of K^+^ in plants and inhibit the various biological activities in which K^+^ participates. Maintaining the homeostasis of ions and water in plants is an important defense strategy against salt stress. Genes such as high-affinity K^+^ transporter 1 (*MdHKT1*), *MdATG10*, sodium/hydrogen exchanger 2-like (*MdNHX8*), nucleobase-ascorbate transporter 7 (*MdNAT7*), and vacuolar H^+^-pyrophosphatase 1 (*MdVHP1*) are involved in ion transport [75,76,77,78,79]. After being sensed by the primary signal receptors on cell membranes, salt stress signals first change the concentration of Ca^2+^ in the cytoplasm through the salt overly sensitive (SOS) signal transduction pathway and calcium-dependent protein kinase (CDPK) cascade reaction pathway. Under salt stress, Na^+^ passes through the receptor channels on the cell membrane such that the signal is transmitted into the cells to increase the Ca^2+^ concentration; in tun, increased Ca^2+^ concentrations promote the interaction of SOS3 and SOS2 to activate kinases, which then activates downstream functional proteins [80]. MdSOS2 is highly homologous to AtSOS2, which improves the salt tolerance of transgenic plants and interacts with MdSOS3, indicating that the SOS pathway plays an important role in the genus *Malus* [81,82]. The CDPK gene *MdCPK1a* in apple is expressed in the nucleus, and its encoded protein is located in the cell membrane. The ectopic expression of *MdCPK1a* in model organisms results in enhanced resistance to oxidative stress and salt stress [83]. The expression of *MdPAT16* (encoding an S-acyltransferase) is induced by salt stress, and the overexpression of this gene promotes the accumulation of soluble sugars. MdPAT16 interacts with MdCBL1 to increase the stability of MdCBL1, which reveals the role of the MdPATs16-MdCBL1-MdCIPK13-MdSUT2.2 pathway in the apple salt stress response [84] (Figure 2). In addition to Ca^2+^-dependent signal transduction pathways, Ca^2+^-independent pathways, such as the MAPK signaling cascade pathway, also play an important role in salt stress signal transduction. The MAPK cascade consists of the following three protein kinases: MAPKKK (MEKK), MAPKK (MEK or MKK), and MAPK. When a cell responds to developmental signals or environmental stress, components of the MAPK cascade are sequentially activated by phosphorylation, and the target protein is phosphorylated by the activated MAPK, which alters the activity of the target protein and transmits the signal. In summary, one hundred and twenty MAPKKK family members, nine MAPKK family members, and twenty-six MAPK family members have been identified in apple. The expression of most MAPK genes was shown to be upregulated in response to salt stress [69,85]. MKK2 is specifically activated by cold and salt stress, and the stress-induced MAPK kinase MEKK1, which directly targets MPK4 and MPK6, and its downstream transcriptional regulation, signal transduction, cell defense activity and stress-related metabolic activity alter the expression of 152 genes. These results directly show that the MAP kinase signaling cascade is involved in the salt stress tolerance of plants [86].

Cold stress mainly includes chilling stress (0–15 °C) and freezing stress (<0 °C) and is one of the main environmental factors that seriously affects plant growth and crop productivity [87]. In the long-term evolutionary process, apple plants evolved a series of mechanisms that enable them to adapt to cold stress at the physiological and molecular levels. After the cold signal is sensed by plant cells, the cold-induced increase in the cytosolic Ca^2+^ concentration is achieved by Ca^2+^ channels that are sensitive to mechanical tension on the membrane or activated by ligands [88,89]. In addition to Ca^2+^, when plants are damaged by stress, the contents of hormones such as ethylene, jasmonic acid (JA) and methyl jasmonate (MeJA) (as signaling molecules) increase significantly to subsequently induce the expression of genes related to stress resistance [90,91,92,93]. The signal generated by cold stress is transmitted basipetally, which causes a change in the expression of COR genes, enhancing plant cold tolerance. To date, many cold acclimation-related genes have been screened, cloned and identified, and the CBF-dependent pathway is reported to be an important and relatively clear cold stress response pathway in plants [94,95]. This pathway is mediated by ICE-CBF-COR signaling, which is involved in cold perception, signal transduction, gene expression, and other processes. The ICE-CBF-COR pathway enhances the cold resistance of plants and acts as a molecular switch [96]. The apple bHLH gene MdCIbHLH1, which has a bHLH 1–type domain, specifically binds to the CBF3 promoter and initiates the transcription of CBF3 [97]. MYB88/124 and other TFs enhance the cold tolerance of apple by inducing the transcription of genes related to the CBF pathway [98,99,100,101,102]. Ubiquitinated E3 ligase specifically recognizes and degrade MdBBX37, increases the expression of CBF1–4, and regulates the cold resistance of plants [98] (Figure 1).

### 2.3. Mineral Nutrient Regulation of the Apple Root System and Design of the Ideal Root System Architecture

Plant root system growth and morphological changes proceed smoothly only in the presence of a timely supply of various nutrients in appropriate proportions. When a certain element is missing or the proportion of an element is imbalanced, the absorption of the rootstock is often hindered, affecting the root system architecture. Plants adapt to the immobility and uneven distribution of nutrient elements in the soil by changing the shape and configuration of their roots, thereby increasing the roots’ ability to absorb nutrients in the topsoil [103,104]. Throughout their biological evolution, plants have evolved complex molecular mechanisms to respond to changes in elements in the soil and adjust the stability of their own element levels in a timely manner. An understanding of the molecular mechanism by which apple responds to nutrient element concentrations and regulates the element balance not only has biological research importance but also facilitates the development of molecular biology methods to select apple rootstocks that efficiently utilize elements, laying a foundation for precision breeding.

Nitrogen (N) is one of the most important elements for plant growth and mainly exists as NO_3_^−^ and NH_4_^+^ in botanical organisms. Analyzing the molecular mechanisms and the regulatory network of key genes by regulating the absorption, utilization and transformation of NO_3_^−^ will provide an important theoretical basis for improving the use of plant-available nitrogen. The NRT1 family of proteins belongs to the NRT/PTR (NPF) family [105,106] (Figure 2). All members of this family have been reported to be able to transport NO_3_^−^. The phosphorylation of NRT1.1 (which encodes an NRT1 protein) is involved in adapting to changes in the external NO_3_^−^ concentration [107,108,109,110]. NRT2.1, a member of the NRT2 cotransporter superfamily, regulates lateral root development and participates in the ethylene signaling pathway in the absence of NO_3_^−^ [111,112,113,114,115]. The protein kinase CIPK8 is induced by NO_3_^−^ in a relatively short time. When the external NO_3_^−^ concentration is high, CIPK regulates the expression of the NRT1.1 and NRT2.1 genes, indicating that CIPK8 is a positive regulator when the NO_3_^−^ concentration is sufficient [116,117,118]. NIN-LIKE PROTEIN 7 (NLP7) interacts with Ca^2+^-sensor protein kinase 10 (CPK10) to jointly regulate the absorption of NO_3_^−^ [119,120,121,122,123]. The high-affinity nitrate transporter NRT2.4 not only regulates nitrate absorption but also regulates nitrate transport [124]. Chloride channel proteins (CLCs) also play important roles in the absorption of nitrate. In the *clca* mutant, the accumulation of NO_3_^−^ is 50% lower than that of the wild type. CLCa is an important component of the pathway regulating NO_3_^−^ accumulation [125,126,127,128]. The ectopic expression of *MhNCED3* in *Arabidopsis* upregulates the expression of *AtCLC*, implying that the former plays a role in nitrate regulation. In addition, ammonium is an important nitrogen source for plant growth. Five members of the ammonium transporter AMT1 subfamily have been identified in *Malus robusta Rehd* rootstock. The AMT1 protein complements the ammonium ion absorption process in an *amt*-deficient mutant yeast system [129] (Figure 2).

Phosphorus (P) is another macronutrient element necessary for plant growth and development, mainly because it plays vital roles in the processes of energy metabolism, sugar metabolism, enzymatic reactions, and photosynthesis. Phosphate, the main carrier of phosphorus, easily binds to metal ions in the soil to form precipitates [130]. The content of available phosphorus in soil is low, which may result in nutrient stress. Many genes that are specifically expressed in response to phosphorus deficiency, including high-affinity phosphorus transporters, organic acid secretion-related genes, acid phosphatase-related genes, and members of the TPSI1/Mt4 gene family, have evolved throughout evolution [131] (Figure 2). Members of the phosphate transporter family in apple have also been identified. Thirty-seven genes are induced by phosphorus, and they play an important role in the response to unfavorable growth environments in apple [132]. Transgenic experiments revealed that TF MdMYB2 regulates the expression of phosphate starvation-induced (PSI) genes, thereby promoting the assimilation and utilization of phosphate [133]. Apple SUMO E3 ligase (MdSIZ1) gene expression is induced by phosphorus deficiency, and the growth of both transgenic *Arabidopsis* and apple calli was shown to increase under phosphorus-deficient conditions, indicating that MdSIZ1 is involved in regulating the response to phosphorus deficiency [134].

Iron (Fe) deficiency in plants is still one of the most common problems in agricultural production. Only a small amount of iron in the soil environment is dissolved in the soil and absorbed by plants, which seriously affects the utilization efficiency of iron by plant roots. After iron is reduced from Fe^3+^ to Fe^2+^ in the roots, it is transported into the cells by iron transporters on the cell membrane. Iron-regulated transporter 1 (IRT1) is the main root transporter that absorbs iron from the soil [135,136] (Figure 2). Fe^3+^ forms a metal ion chelate with citric acid, which is then transported to various organs, aided by iron ion transporters [137,138,139]. The v-myb avian myeloblastosis viral oncogene homolog (*Md*MYB58) and multidrug and toxin efflux (*Md*MATE43) proteins interact to regulate Fe homeostasis, and the ectopic expression of their genes in *Arabidopsis* calli leads to the accumulation of Fe [140]. SUMO E3 ligase (*Md*SIZ1) in apple affects the ubiquitin-like modifier (SUMO) modification in MdbHLH104 under iron-deficient conditions to promote both the transcription of downstream genes and the reduction and absorption of iron ions [141,142]. According to related reports, red light activates the key TF ELONGATED HYPOCOTYL 5 (HY5) through the photoreceptor PhyB, further regulating the expression of key factors involved in root iron absorption and promoting iron absorption [143,144,145,146]. In summary, an understanding of the functions of genes in model organisms will facilitate the molecular breeding of apple rootstocks.

### 2.4. Root Architecture and Hormone Regulation

Phytohormones, which are trace amounts of active substances produced in plants, are transferred from biosynthesis sites to active sites. Phytohormones independently regulate seed germination, growth and development, embryonic development, root system architecture, and other biological life cycle processes. At present, substantial progress has been achieved in the field of apple plant hormone research. Moreover, many hormone metabolism and signal transduction pathways have been discovered, including the molecular mechanisms and biological effects of various hormones, such as auxin [indole-3-acetic acid (IAA)], cytokinin (CK), and N-terminal signal peptides.

The earliest discovered plant hormone, auxin, plays an extremely critical role in the complex hormone regulation of plant root development networks. Genes related to auxin synthesis, transport, and signal transduction affect root architecture to varying degrees. Auxin signaling is an important component required for the normal growth and development of plants and mainly depends on auxin response factor (ARF)-Aux/IAA interactions. ARFs are TFs that regulate auxin responses in plants. In apple, 31 putative ARF genes have been identified and divided into the following three subfamilies: I, II, and III [147]. Furthermore, the function of MdARF13 was investigated, and the MdARF13 protein was found to interact with the Aux/IAA repressor MdIAA121 through its C-terminal dimerization domain [148]. The degradation of MdIAAs mediated by TRANSPORT INHIBITOR RESISTANT 1 (MdTIR1) is essential for apple auxin signal transduction and root-growth regulation [149]. A wild apple (*Malus sieversii* Roem) GRETCHEN HAGEN 3 (GH3) gene, MsGH3.5, encodes an IAA-amido synthetase that affects apple growth and development by modulating auxin levels and signaling pathways (Figure 3). These findings provide insights into the interaction between the auxin and CK pathways and might have substantial implications for efforts to improve apple root architecture [150,151]. In apple plants, auxin is transported in the following two ways: long-distance vascular transport and short-range active transport, which requires carriers. The latter plays a key role in the asymmetric distribution of auxin, which is also called polar auxin transport. Polar auxin transport depends on PIN-FORMED (PIN) family proteins, AUXIN1/LIKE-AUX1 (AUX/LAX) family proteins, and ATP-binding cassette subfamily B (ABCB) proteins. IAA polar transport largely depends on PIN auxin efflux carriers. [152]. FOUR LIPS (MdFLP) directly binds to the promoters of two auxin efflux carriers, MdPIN3 and MdPIN10, which are involved in auxin transport, activate their transcription and thereby promote the development of adventitious roots in self-rooting apple stocks [153]. ABCB1 and ABCB19 are two well-studied auxin efflux carrier genes and may play a role in auxin efflux [154]. Auxin alters plant growth and the development of many processes, and plays an important regulatory role in embryonic development, organogenesis, root anisotropic growth, and other processes.

Regulatory interactions between auxin and cytokinin (CTK) pathways control lateral root development. CTKs, which are generally produced in the roots, constitute a class of substances that promote cytokinesis, differentiation, and the growth of various tissues. CTK synergy with auxin regulates plant cell growth and the development of plant hormones. CTK levels are maintained mainly by CTK biosynthesis (via isopentenyl transferase [IPT]) and degradation [via cytokinin dehydrogenase (CKX)] genes in plants [155] (Figure 3). Isopentenyltransferase5b (*IPT5b*), a gene involved in CTK biosynthesis and catabolism, works to maintain a high CTK content in apple rootstock roots. Under salt stress, apple roots maintain high CTK levels, as *IPT5b* expression is not inhibited by proline due to the deletion of the proline response element (ProRE), leading to improved root cell division activity [156] (Figure 3). In addition, *IPT5b* expression increases when M9 rootstocks are treated with 5-azacytidine (5-azaC), a methylation blocker, indicating that methylation levels influence *IPT5b* expression; therefore, CTK synthesis is affected [157]. Thr overexpression of *MsDREB6.2* was shown to increase the expression of a key CTK catabolism gene, *MdCKX4a*, which led to a significant reduction in endogenous CTK levels and caused a decrease in the shoot/root ratio in transgenic apple plants [158]. CTKs interact extensively with other hormones, but their responses to different plant hormone signals and their potential tissue and cell specificity remain unclear. Combining research on mutants with gain and loss of gene function through corresponding specific inhibitors and various omics research, including transcriptomics, proteomics, metabolomics, and phenomics, may help to explain the relationships between key hormone-related genes and plant root phenotypes.

In addition, active substances have been shown to regulate the development of plant roots. Small peptides constitute an important component of interactions and signaling between cells, and their biological functions and underlying mechanism have become popular and timely research topics in plant research in recent years. Since the discovery of the first plant peptide hormone and after years of exploration and research, peptide hormones have been gradually valued and widely studied because of their important role in regulating plant growth and development and adapting to environmental conditions. C-TERMINALLY ENCODED PEPTIDE (CEP) exerts an inhibitory effect on the root growth of apple seedlings, indicating that MdCEP1 negatively regulates root development [159]. The CLE16, CLE17, and CLE27 members of the CLAVATA 3/EMBRYO SURROUNDING REGION-RELATED (CLE) gene family play largely redundant roles in the *Arabidopsis* root apical meristem and/or regulate meristem activity under only specific environmental conditions [160] (Figure 3). As an intermediate regulator, RECEPTOR-LIKE PROTEIN 44 (RLP44) participates in PSK-mediated signaling pathways by regulating the interaction between receptors and coreceptors and is also regulated by the BRASSINOSTEROID INSENSITIVE 1 (BRI1) feedback mechanism, thereby regulating the cell fate of plant xylem and thus the root system [161]. Furthermore, β-carotene-derived retinoid plays important roles in circadian clock oscillation, neurogenesis, and vasculature development in animal cells. Recent studies found that retinol also mediates circadian clock oscillations and regulates the development of plant lateral roots in plant cells [162].

The involvement of plant hormones in the formation of adventitious roots is a complicated process. To date, few studies have focused on the interaction of the abovementioned plant hormones in inducing adventitious root formation, and most of these studies are still at the level of traditional plant physiology. Therefore, with the help of mutants with defects in lateral root or adventitious root development, especially mutants with defects in hormone signal transduction and hormone synthesis, current research may apply the rational use of molecular biology and genetics combined with appropriate model plant species to study the development of adventitious roots. This review therefore recommends undertaking future studies on the molecular mechanism and signaling network of hormones involved in regulating the development of adventitious roots and contributes to precision breeding.

## 3. Conclusions and Future Prospects

Currently, uncultivated wild apple, a wild relative species of the same genus as cultivated apple rootstocks, is not only an important germplasm resource that must be preserved for cultivating new varieties of rootstocks but also a global strategic reserve resource. The abundance of wild resources of apple rootstocks and individual domesticated varieties provide both direction and development opportunities for the advancement of the apple industry. Phenotypic variability during evolution may be a result of environmental factors such as environmental changes, resulting in the occurrence of chromosomal timing. Phenotypic differences within the same breed may be due to changes in a single base or a single gene. In addition, transposon sequences are widely present in plant genomes and play an important role in crop domestication. In the cabbage genome, the transposon content is as high as 50%. Studies of transposon insertions reveal sources of phenotypic variation [163]. Gene loss due to natural LoF (loss-of-function) mutations, which are ubiquitous in the genomes of various organisms, has been somewhat overlooked. With the development of sequencing technology, many genomes have been sequenced in different species and used to study the variants contributing to variations in gene loss [164]. SoGV (somatic genetic variation) produces genetic variations late in ontogeny and is inherited in gametes, and sexual and asexual processes provide pathways for interdependent variant variation, affecting genetic load accumulation and molecular evolution in the integrated asexual/sexual life cycle [165]. However, previous methods that involve screening hundreds of plant phenotypes indeed proved to be too costly. From the perspective of rootstocks, research on both root traits and root architecture is insufficient. In fact, a large number of target sites pinpointing genetic resources have not yet been developed. Moreover, the apple transgenic transformation system is still in its infancy, and a large number of key trait genes cannot be used in production.

Current solutions include (1) using a wide range of wild resources to conduct a wider range of trait statistical analyses; (2) exploiting computer learning to screen excellent apple chassis materials (which might be the best solution available); and (3) using models such as genotype-to-phenotype prediction or whole-genome selection prediction to select accessions based on genotypes. In future breeding processes, the material composition, line selection, assembly, testing, promotion, and many other components might be determined using a series of models designed for breeding decision making to assist or even replace manual decision making, ultimately achieving data-driven intelligent design-based breeding. Gene-editing technologies using new methods, such as clustered, regularly interspaced and short palindromic repeats (CRISPR)/CRISPR-associated 9 (Cas9), have accelerated the innovation of new crop plant breeding materials and the cultivation of new varieties [165,166]. To date, breakthroughs in gene editing technology innovations for more than 20 crop species have been reported, including corn, rice, wheat, and soybean. A number of commercially valuable varieties have been cultivated and gradually promoted for breeding. Based on gene-editing technology, the development of polyploidy technology to correct the heterosis of apomictic reproduction might also be applied to rootstock breeding.

In this review, we discussed the genetic diversity of apple rootstocks among varieties, information on genome sequencing and provided a functional annotation of different varieties of apple, and information on the development of molecular markers and functional genes. According to the current literature, future genome studies should increase the density of genetic maps and improve molecular markers, which is more important for researching the genetic breeding of apple combined with the phenotype. Conventional breeding and phenotypic breeding are now being gradually replaced by molecular breeding, which will be replaced by intelligent breeding in the future. In this sense, methods to design intelligent farming systems for apple warrant further investigation; apple germplasm resources constitute the genetic material basis for cultivating new varieties or types of apples and provide usable rootstocks for production. Therefore, they are very important for the development of apple production. We propose that an apple germplasm bank and genome should therefore be improved and established. The use of genomic information and key genes for classical breeding combined with germplasm resources for germplasm innovation is helpful. Promoting the industrialization of classical breeding with a scientific and rigorous attitude is also important. At present, most global agricultural development must focus on quantitative needs, strengthening crop genetic breeding efforts, and promoting the development of the modern agricultural industry, which is an important research direction to ensure the security of global agriculture.

## Figures and Tables

**Figure 1 plants-11-01408-f001:**
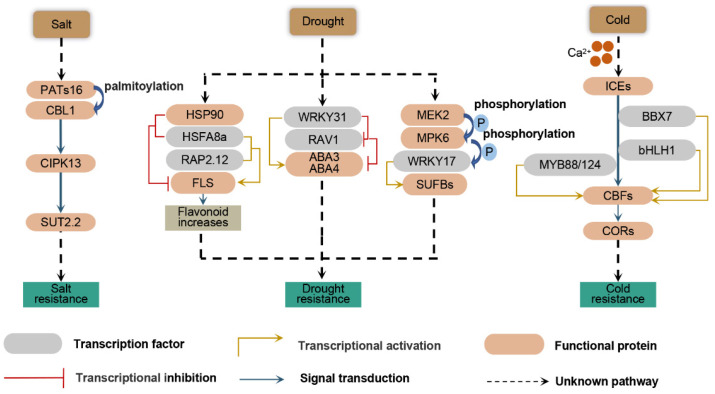
Summary of the role of the control network and signal transduction-dependent pathways in responses to biotic stress and abiotic stress in apple. Under salt stress, MdPAT16 participates in the regulation of the MdCIPK13-regulated salt stress response and sugar accumulation through the palmitoylation of MdCBL1. Under drought stress, the MdHSP90-MdHSFA8a complex dissociates, and the released MdHSFA8a interacts with the AP2/ERF family transcription factor MdRAP2.12, thereby activating the activities of the related structural genes in the downstream flavonoid pathway. MdWRKY31 participates in the ABA signaling pathway by inhibiting the transcriptional activity of MdRAV1 to enhance the transcriptional activity of MdABI3 and MdABI4. The MdMEK2-MdMPK6-MdWRKY17-MdSUFB signaling pathway stabilizes the chlorophyll content in the leaves of apple plants under moderate drought stress. Cold stress mediated by ICE-CBF-COR is induced by transcription factors such as BBX, MYB, and bHLH. PAT16: protein palmitoyl transferase 16; CBL1: calcineurin B-like 1; CIPK13: CBL-interacting protein kinase; SUT2.2: sucrose transporter 2.2; HSP90: heat shock protein; HSFA8a: heat shock factor A8a; RAP2.12: RELATED TO APETALA 2.12; FLS: flavonol synthase; RAV1: related to ABI3/VP1 1; ABA3, 4: molybdenum cofactor sulfurase 3,4; MEK2: mitogen-activated protein kinase 2; MPK6: mitogen-activated protein kinase 6; SUFB: sulfur mobilization B; ICE: inducer of CBF expression; BBX7: B-Box protein 7; bHLH1: basic helix-loop-helix 1; MYB88/124: V-myb avian myeloblastosis viral oncogene homolog 88/124; CBFs: C-repeat binding factors; CORs: cold-responsive genes.

**Figure 2 plants-11-01408-f002:**
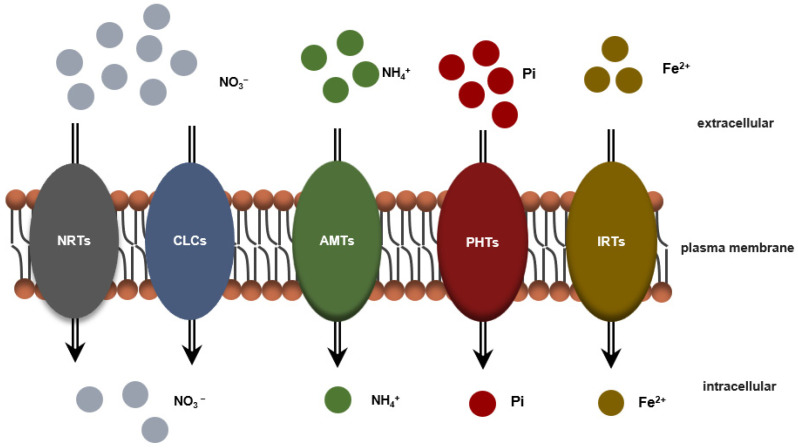
Working model for the direct target proteins regulating nutrient absorption in apple. Macro- and micronutrients such as NO_3_^−^, NH_4_^+^, Pi, and Fe^2+^ are transported into plants through membrane transporters such as NRTs, AMT, CLCs, PHTs, and IRTs. NRTs: nitrate transporters; CLCs: chloride channel proteins; AMTs: ammonium transporters; PHTs: phosphate transporters; IRTs: iron-regulated transporters.

**Figure 3 plants-11-01408-f003:**
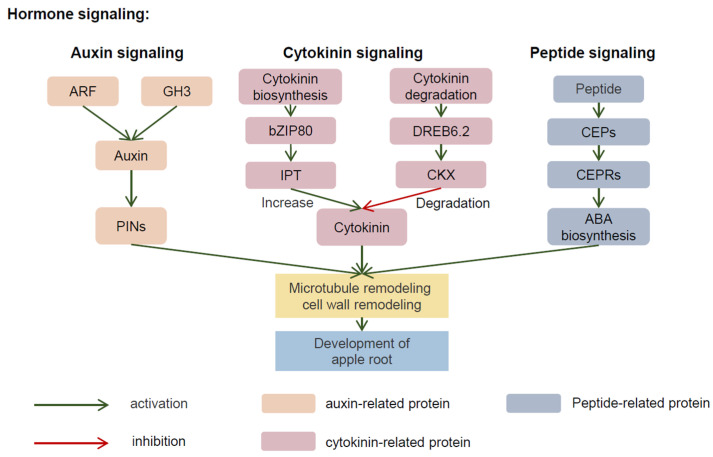
The phases of the mechanism regulating the root system configuration. The signaling component is activated by hormones. In apple, ARF and GH3 are involved in the synthesis of auxin, and the efflux of auxin is inseparable from PINs. CK levels are maintained mainly by CK biosynthesis (isopentenyl transferase, IPT) and degradation (cytokinin dehydrogenase, CKX) genes. Some CLEs have the ability to promote ABA synthesis, which is transferred from the root long distance to the leaves to bind receptor proteins, resulting in an increased ABA concentration and participation in the appropriate function. ARF: auxin response factor; GH3: Gretchen Hagen 3; PINs: PIN-FORMED (PIN) auxin transporters; bZIP80: Basic region/leucine zipper motif 80; IPT: isopentenyl transferase; CKX: cytokinin dehydrogenase; CEPs: C-TERMINALLY ENCODED PEPTIDE; CEPRs: CEP receptors; ABA: abscisic acid.

## Data Availability

Not applicable.

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
