# Peer review of "Root Breeding in the Post-Genomics Era: From Concept to Practice in Apple"

_plants, 2022, doi:10.3390/plants11111408_

Round 1
Reviewer 1 Report
Dear Authors
The work entitled "Root Genetics and Root System Architecture: A Rootstock De- 2 sign and Breeding Support" presents a problem important from the point of view of agricultural practice to be solved in the future and therefore it deserves to be published in your journal. It does not mean, however, that the work cannot be better, on a higher scientific level. Here are the details:
- The summary of the work does not contain the most principal elements, i.e., the purpose of the work, the shortened methodology of the research and the general conclusion.
- The introduction to the thesis is sufficiently extensive, but it does not contain a clear aim of the work and there is no alternative research hypothesis in view of the null hypothesis. The verification of this hypothesis should be included later in the paper.
- There is no chapter "Methodology", in which the authors should describe how they collected data for work, how they analyzed the collected material, what tools they used for this purpose.
- The rest of the work includes a discussion of the results and a discussion of the issues presented, broken down into well-organized chapters and subchapters. What I miss here is a detailed discussion of phenotypic variability, which is a critical issue in plant breeding.
- There is no general summary or conclusion, although there is talk of the future in tree apple breeding.
Author Response
Response to Reviewer 1 Comments
Point 1: The summary of the work does not contain the most principal elements, i.e., the purpose of the work, the shortened methodology of the research and the general conclusion.
Response 1: Thank you for your comments on improving our entire article. We have made major revisions to the paper by improving the descriptions of the goals of the paper and our views. Apple rootstocks are the key to high-quality and high-yield apple trees, but their long breeding cycle and difficulty in root character research limit the variety and precise creation of rootstock varieties. Therefore, we summarize the research progress of apple genetic evolution and molecular mechanisms, with the aim of using complex reproductive methods for apple, such as self-compatibility, self-incompatibility, and apomixis, to improve traditional breeding. More importantly, we can learn from root regulation genes to achieve breakthroughs in root nutrient absorption, root resistance and root configuration regulation of rootstocks. However, an important link between traditional breeding and modern precision molecular breeding is an efficient genetic transformation system without genotype barriers, but it is still lacking. Therefore, based on your guidance, we have added the following objective: to explore new methods of rootstock genetic improvement combined with traditional breeding. Based on the overall modification, we proposed the conclusions listed below. A wide range of wild resources and self-incompatibility characteristics should be used to establish a variety of distant hybrid populations under natural conditions, create multicombination hybrid offspring, achieve the diversified allocation of apple genetic resources; and overcome the limitation of the narrow genetic background of existing rootstocks. At the same time, a genetic transformation system and gene editing system that are not restricted by genotype should be established to accurately improve rootstock nutrient absorption, resistance and ideal root type regulation. We integrated and revised the summary section. “This paper describes ideas for the precise breeding and domestication of wild apple resources and application of key genes. The primary goal of this research is to combine the existing rootstock resources with molecular breeding and summarize the methods of precision breeding.” was added on page 1, lines 20-25.
Point 2: The introduction to the thesis is sufficiently extensive, but it does not contain a clear aim of the work and there is no alternative research hypothesis in view of the null hypothesis. The verification of this hypothesis should be included later in the paper.
Response 2: Thank you for your valuable guidance. The apple rootstock breeding cycle is long, rootstock root system research is difficult, and systematic research on rootstock re-sources, such as the extensive adaptability of resources, grafting affinity between re-sources and multiple varieties, and genetic characteristics of diverse combinations of resources, is still lacking. In terms of molecular mechanisms, due to the limitation of species and functional redundancy of alleles, sequences and functional diversity in re-sources are relatively lacking. Therefore, we did not propose a clear working goal in the previous version of the paper. Based on your prompt, we have revised the whole manuscript and tried to describe our ideas for variety improvement. The combination of traditional breeding and modern breeding uses a wide range of wild resources and self-incompatibility characteristics to establish a variety of distant hybrid populations under natural conditions, create multicombination hybrid offspring, and achieve the diversified allocation of apple genetic resources. At the same time, a genetic transformation system and gene editing system that are not restricted by genotype should be established to accurately improve rootstock nutrient absorption, resistance and ideal root type regulation. We have revised that sentence as follows to present a clear aim on page 2, lines 47-52: “However, the roots of rootstocks are buried deep underground, and many limitations are encountered when only relying on past experience planting to select and breed rootstock varieties with excellent varieties, strong stress resistance and strong root system configurations. Therefore, the use of modern biological technology to improve breeding and achieve the purpose of the precise selection of apple rootstocks is an urgent need.”
Point 3: There is no chapter "Methodology", in which the authors should describe how they collected data for work, how they analyzed the collected material, what tools they used for this purpose.
Response 3: Thank you for your advice, and we describe the methodology on page 2, lines 92–99: “In this review, several databases, including Elsevier, PubMed, SpringerLink, and Google Scholar, were used for the literature search. A combination of keywords, such as apple germplasm resources, genomics, precise breeding, abiotic stress, biotic stress, drought, salt, temperature stress, nutrient regulation, and hormone regulation, were used as search terms. Publications from 2010-2021 were preferred, but this range was extended in some cases. After searching the literature, we read each paper carefully and thoroughly to exclude those that did not meet the inclusion criteria. We initially selected 160 papers related to these targets.”
Point 4: The rest of the work includes a discussion of the results and a discussion of the issues presented, broken down into well-organized chapters and subchapters. What I miss here is a detailed discussion of phenotypic variability, which is a critical issue in plant breeding.
Response 4: Thank you for bringing this issue to our attention. We neglected this important issue in plant breeding in our discussion, and we have added conclusions. “Phenotypic varia-bility during evolution may be due to environmental factors such as environmental changes, resulting in the occurrence of chromosomal timing. Phenotypic differences within the same breed may be due to changes in a single base or a single gene. In addition, transposon sequences are widely present in plant genomes and play an important role in crop domestication. In the cabbage genome, the transposon content is as high as 50%. Studies of transposon insertions reveal sources of phenotypic variation [163]. Gene loss due to natural LoF (loss-of-function) mutations, which are ubiquitous in the genomes of various organisms, has been somewhat overlooked. With the development of sequencing technology, many genomes have been sequenced in different species and used to study the variants contributing to the variation in gene loss [164]. SoGV (somatic genetic variation) produces genetic variation late in ontogeny and is inherited in gam-etes, and sexual and asexual processes provide pathways for interdependent variant variation, affecting genetic load accumulation and molecular evolution in the integrated asexual/sexual life cycle [165].” Please see page 14, lines 609-624.There is no general summary or conclusion, although there is talk of the future in tree apple breeding.
Point 5: There is no general summary or conclusion, although there is talk of the future in tree apple breeding.
Response 5: Thank you for your valuable guidance. According to your suggestion, we have further developed our conclusions and revised the original text as described below. A wide range of wild resources and self-incompatibility characteristics should be used to establish a variety of distant hybrid populations under natural conditions, create multi-combination hybrid offspring, achieve the diversified allocation of apple genetic re-sources, and overcome the narrow genetic background of existing rootstocks. Although breakthroughs in molecular mechanisms have been reported, direct application of existing genetic resources to apple breeding is impossible due to the functional redundancy of alleles, the species diversity of sequences, and the lack of a good transformation system. At the same time, the systematic molecular mechanism has not been completely elucidated, such as the nutritional regulation of root system configuration and the theoretical basis for the precise regulation of root systems by components such as small peptides and hormones. Therefore, the combination of traditional breeding and modern breeding using a wide range of wild resources and self-incompatibility characteristics establishes a variety of distant hybrid populations under natural conditions, creates multicombination hybrid offspring, and achieves the diversified allocation of apple genetic resources. At the same time, a genetic transformation system and gene editing system that are not restricted by genotype should be established to accurately improve rootstock nutrient absorption, resistance and ideal root type regulation. We revised the text on page 14, lines 645-650 as follows: “In this review, we have discussed the genetic diversity of apple rootstocks among varieties, information on genome sequencing and functional annotation of different varieties of apple, and information on the development of molecular markers and functional genes. According to the current literature, future genome studies should increase the density of genetic maps and im-prove molecular markers, which is more important for researching the genetic breeding of apple combined with the phenotype.”

Reviewer 2 Report
Dear Authors
A minor however critical revision for English language is required throughout the manuscript. The manuscript is well written with all essential information, infographics and literature.
Author Response
Response to Reviewer 2 Comments
Point 1: A minor however critical revision for English language is required throughout the manuscript. The manuscript is well written with all essential information, infographics and literature.
Response 1: According to your comments,we revised and polished the article with the help from professional team of American Journal Experts.

Reviewer 3 Report
Comments to the paper: Root Genetics and Root System Aarchitecture: A Rootstock Design and Breeding Support by Zhou Zhou et al.
The paper contains review of the available knowledge regarding molecular studies on the root system architecture in the genus Malus. In general, the paper is well written and includes wide range of literature sources. However, in my opinion it requires quite a few serious linguistic as well as other corrections before it could be considered for publication. Please see more detailed comments presented below.
- The title is misleading, as it gives impression that the paper reports results of an experimental work. It should clearly indicate that it represents a review. Furthermore, it should indicate the genus of the plant (both Latin and common) described in the text.
- The authors should pay great attention to both grammar and spelling! I was rather surprised to find spelling mistakes in the title “Aarchitecture” and the authors’ names “wang”. If I were to list all similar mistakes, it would be very long. In the following points I will give few more examples, but I believe that as reviewer I am not supposed to provide full language revision.
- Page 1, lines 29 – 30. I believe that this sentence is an overstatement. Humans can certainly survive without apples.
- Page 1, lines 32 – 36. The authors missed the fact that when plural form “Rootstocks” of a noun is used the following verbs should not end with “s”.
- Page 1, lines 36 – 37. No need to repeat phrase “root system”.
- Page 1, line 39. The authors should introduce Latin name “Malus” before referring to the common name “apple”. This problem occurs throughout the whole text.
- Page 1, lines 43 – 45. Wrong grammar.
- Page 2, line 51. I do not understand how “molecular breeding” can overcome “species isolation”.
- Page 2, lines 74 – 76. When numbers in the range 0 – 10 are used they should be spelled out: six, eight etc.
- Page 2, lines 77 – 79. I do not understand what the authors wanted to say. Furthermore, the word “Begonia” should be in italics.
- Page 2, lines 90 – 91. I think, this sentence is redundant.
- Page 3, line 100. What do the symbols “M” and “MM” stand for?
- Page 3, line 103. What is the scientific name of the pathogen causing “neck rot”?
- Page 3, line 119. The expression “has begun to start” is awkward. Better write: “has begun” or “has started”.
- Page 3, line 132 – 136. Why the length of the genotype promoted research?
- Page 3, lines 138 – 139. It seems that the authors regard terms “molecular evolution” and “microevolution” as synonyms. I disagree.
- Page 4, line 173. Only in apple? I believe that the same can be said about many other domesticated plants.
- Page 6, line 250 and few others. Latin names should be given in italics.
- Page 8, line 349. What exactly do you mean by “biological organisms”?
- Page 8, lines 349 – 352. Wrong grammar.
- Page 8, lines 375 – 376. Missing reference.
- Page 11, lines 488 – 489. Wrong grammar.
- Page 11, lines 503 – 507. You did not apply anything. You just reviewed existing literature on the subject.
- Please check the list of References! It contains errors e.g., page 12, line 568 and many others. Please use EndNote or similar software to fix these errors.

Author Response
Response to Reviewer 3 Comments
Point 1: The title is misleading, as it gives impression that the paper reports results of an experimental work. It should clearly indicate that it represents a review. Furthermore, it should indicate the genus of the plant (both Latin and common) described in the text.
Response 1: According to the comments, and the title of this manuscript has been modified.
Point 2: The authors should pay great attention to both grammar and spelling! I was rather surprised to find spelling mistakes in the title “Aarchitecture” and the authors’ names “wang”. If I were to list all similar mistakes, it would be very long. In the following points I will give few more examples, but I believe that as reviewer I am not supposed to provide full language revision.
Response 2: According to the comments,we have revised and polished the article with the help from professional team of American Journal Experts.
Point 3: Page 1, lines 29 – 30. I believe that this sentence is an overstatement. Humans can certainly survive without apples.
Response 3: According to the suggestion, this sentence has been modified on page 1, line 38.
Point 4: Page 1, lines 32 – 36. The authors missed the fact that when plural form “Rootstocks” of a noun is used the following verbs should not end with “s”.
Response 4: Thank you for your comments, the errors have been corrected accordingly.
Point 5: Page 1, line 39. The authors should introduce Latin name “Malus” before referring to the common name “apple”. This problem occurs throughout the whole text.
Response 5: According to the comments. We have added the Latin names when they first appeared, and we unified expressions throughout the text.
Point 6: Page 1, lines 43 – 45. Wrong grammar.
Response 6: We are sorry for this careless mistake. the description of line 61-66 has been revised.
Point 7: Page 2, line 51. I do not understand how “molecular breeding” can overcome “species isolation”.
Response 7: Thank you very much for reminding us that we used inappropriate terminology. In the revised manuscript, the phrase “specific isolation” has been replaced by “outbreeding incompatibility” in line 77 on page 2.
Point 8: Page 2, lines 74 – 76. When numbers in the range 0 – 10 are used they should be spelled out: six, eight etc.
Response 8: According to the suggestion, we have modified this error accordingly on page 2, lines 112-114.
Point 9: Page 2, lines 77 – 79. I do not understand what the authors wanted to say. Furthermore, the word “Begonia” should be in italics.
Response 9: The sentence in Line 114-118 has been replaced by “Phenotypic or genetic differences between the same species are often referred to as varieties”.
Point 10: Page 2, lines 90 – 91. I think, this sentence is redundant.
Response 10: According to the suggestion, this sentence have been deleted in the revised manscript.
Point 11: Page 3, line 100. What do the symbols “M” and “MM” stand for?
Response 11: Thank you for your comments, “M” and “MM” are the names of two apple rootstock varieties. In the revised manuscript, we have added one sentence “apple rootstock varieties that are widely used in production” to explain this on page 3, line 139.
Point 12: Page 3, line 103. What is the scientific name of the pathogen causing “neck rot”?
Response 12: The scientific name of the pathogen causing “neck rot” is Phytophthora cactorum (Leb. et Cohn.) Schrot, which has been added in line 143 in the revised manuscript.
Point 13: Page 3, line 119. The expression “has begun to start” is awkward. Better write: “has begun” or “has started”.
Response13: According to the suggestion, “has begun to start” has been replaced by “have just begun” in page 4, line 162.
Point 14: Page 3, line 132 – 136. Why the length of the genotype promoted research?
Response 14: Very sorry for this mistake. This sentence has been revised on page 3, line178.
Point 15: Page 3, lines 138 – 139. It seems that the authors regard terms “molecular evolution” and “microevolution” as synonyms. I disagree.
Response 15: According to the comments, the word “microevolution” has been deleted in line 183 on page 3 of the revision manuscript.
Point 16: Page 4, line 173. Only in apple? I believe that the same can be said about many other domesticated plants.
Response 16: We are sorry for this mistake. This sentence has been revised on page 4, line 216.
Point 17: Page 6, line 250 and few others. Latin names should be given in italics.
Response 17: Thank you for your comments and we have modified this error.
Point 18: Page 8, line 349. What exactly do you mean by “biological organisms”?
Response 18: Very sorry for this mistake. The phrase “biological organisms” has been replaced by “botanical organisms” on page 10, line 420.
Point 19: Page 8, lines 349 – 352. Wrong grammar.
Response 19: According to the comments, errors have been modified accordingly in page 8, line 439-442.
Point 20: Page 8, lines 375 – 376. Missing reference.
Response 20: The new references have been added in page 11, line 450 and listed in the reference part.
Point 22: Page 11, lines 503 – 507. You did not apply anything. You just reviewed existing literature on the subject.
Response 22: According to the comments, this sentence has been revised on page 14, line 595 of the revised manuscript.
Point 23: Please check the list of References! It contains errors e.g., page 12, line 568 and many others. Please use EndNote or similar software to fix these errors.
Response 23: According to the suggestion, all the references have been checked and changed using the same format.
